# Strengthening the Trialability for the Intention to Use of mHealth Apps Amidst Pandemic: A Cross-Sectional Study

**DOI:** 10.3390/ijerph19052752

**Published:** 2022-02-27

**Authors:** Munshi Muhammad Abdul Kader Jilani, Md. Moniruzzaman, Mouri Dey, Edris Alam, Md. Aftab Uddin

**Affiliations:** 1Department of Human Resource Management, Bangladesh Institute of Governance and Management (BIGM), Dhaka 1207, Bangladesh; mmakjilani@bigm.edu.bd (M.M.A.K.J.); md.moniruzzaman@bigm.edu.bd (M.M.); 2Bangladesh Bridge Authority, Ministry of Road Transport and Bridge, Dhaka 1212, Bangladesh; 3Department of Accounting, University of Chittagong, Chattogram 4331, Bangladesh; mouridey@cu.ac.bd; 4Faculty of Resilience, Rabdan Academy, Abu Dhabi P.O. Box 22401, United Arab Emirates; 5Department of Geography and Environmental Studies, University of Chittagong, Chattogram 4331, Bangladesh; 6Department of Human Resource Management, University of Chittagong, Chattogram 4331, Bangladesh

**Keywords:** mHealth, DOI model, Generation Y, COVID-19

## Abstract

Recent advancements in mHealth apps and services have played a vital role in strengthening healthcare services and enabling their accessibility to marginalized people. With the alarming rise in COVID-19 infection rates around the world, there appears to be an urgent call to modernize traditional medical practices to combat the pandemic. This study aims to investigate the key factors influencing the trialability of mHealth apps/services and behavioral intention to adopt mobile health applications. The study also examines the moderating effects of self-discipline motivation, knowledge, and attitude on the relationship between trialability and behavioral intention to use. The deductive reasoning approach was followed in a positivism paradigm. The study used convenience sampling and collected responses from 280 Generation Y participants in Bangladesh. Partial least square-based structural equation modeling was employed. The results revealed that relative advantage (β = 0.229, *p* < 0.05), compatibility (β = 0.232, *p* < 0.05), complexity (β = −0.411, *p* < 0.05), and observability (β = 0.235, *p* < 0.05) of mHealth apps influence the trialability of mHealth apps and services among users. Trialability compatibility (β = 0.425, *p* < 0.05) of mHealth was positively related to the behavioral intention to use these mobile apps. The study found no moderating effects of attitude (β = 0.043, *p* > 0.05) or self-discipline motivation (β = −0.007, *p* > 0.05) on the hypothesized relationships. The empirical findings of this study may facilitate the development, design process, and implementation of mHealth applications with improved features that can lead to high user acceptance among Generation Y during future health crises.

## 1. Introduction

With the rapid advancement of information and communication technology in the startup phase of the industry 4.0 age, mobile applications are becoming more prevalent in a variety of fields that appeal primarily to young people [1,2,3]. The COVID-19 pandemic gives rise to the wide use of health applications, as non-pharmaceutical measures including social distance, isolation, and lockdown are enacted to reduce the number of unnecessary visits of people with compromised immune systems to sensitive locations to contain transmission [4,5,6,7]. These applications enable patients and doctors to stay connected while tracking COVID-19 cases and developing prevention strategies [8,9,10]. 

In recent weeks, young adults in Bangladesh have been infected with SARS-CoV-2 [11]. This population frequents the workplace, congregates in public places, and has a false sense of security, thereby transmitting the disease to the other age groups unwittingly [12]. In densely populated Bangladesh, where there are approximately 3.05 physicians per 10,000 people and 1.07 nurses per 10,000 people, non-therapeutic measures are out of control [13,14]. Thus, mHealth apps supported by smartphone and artificial intelligence technologies offer an alternative solution for healthcare to patients that ensure instantaneous and accessible medical services for contact tracing, health tracking, prescribing treatment, paperless medical documentation, monitoring, and health management for suspected COVID-19 patients [15,16,17,18]. mHealth apps are an innovative healthcare tool with the potential to enhance health communication among patients and clinicians and to improve patients’ health outcomes in a pandemic without physical movements [19,20,21].

The increased use of mHealth technologies has created the potential to access specialist clinical diagnostics and treatment advice [22,23] because it can lower costs, save time, and increase socially distant interaction among patients, nurses, and doctors while providing a quick way to send disease and health-related messages [24,25]. Recently, the Government of Bangladesh, in partnership with the private sector, launched an innovative mHealth app named “Corona Tracer BD” for contact tracing to help citizens prevent the spread of COVID-19 in Bangladesh. Unfortunately, the trialability of the app has completely failed and early adopters uninstalled it after a few weeks due to its ineffectiveness [26]. The main challenges stem from technical, usability, and privacy issues, and requirements reported by some users outweigh potential benefits [27,28,29].

The present study attempts to unearth the predictors of the mHealth apps’ trialability (availing free trial before actual use) and behavioral intention to use (perceived voluntariness of an individual to use) among Generation Y because this generation is liberal, technologically literate, self-expressive, competent, and willing to learn new ways of doing things [30]. Generation Y belongs to the age groups from the early 1980s to early 2000s [31,32]. This group constitutes 39.7 percent of the total population in Bangladesh [33]. Not only are they are exposed to COVID-19 due to their jobs and other business activities, but they also have more expertise and access to modern devices compared to older generations [34]. However, intention to adoption is still slow, and not all Generation Y individuals use their mobile phones to access health services [24]. Now, more than ever during this pandemic, adopting mHealth apps is critical to reducing the risk of contact for both users and health-care providers. In this regard, mHealth trialability is a prerequisite for determining whether mHealth apps meet the adoption criteria of users [35]. mHealth trialability refers to the opportunity to test, evaluate, and experiment with mHealth apps among Generation Y to check their suitability for use based on final adoption criteria [36]. 

Therefore, it is important to explore the underlying prerequisites for the trialability of mHealth apps that influence its adoption among Generation Y. Adoption of mHealth during COVID-19 remains a large challenge for developing countries [33]. Extensive research on the user adoption process is required, especially considering users who have vast knowledge about using mobiles or smartphones [30,37]. Contrarily, limited research was conducted about the predictors affecting the trialability of mHealth apps that led to the behavioral intention to use them. Studies show that individuals having a strong self-discipline motivation and a positive attitude use more health-related services on mHealth platforms [38,39]. However, the effects of self-discipline motivation and attitude toward mHealth apps on the behavioral intention to use have not been well-researched.

This study looked into the connections between Rogers’ perceived characteristics of innovation (i.e., relative advantage, compatibility, complexity, observability, and trialability) and behavioral intention to use mHealth apps among Generation Y using the diffusion of innovation (DOI) theory [35]. The present study used a deductive reasoning approach and collected cross-sectional data to test whether the trialability of mHealth apps, interacting with self-discipline motivation, knowledge, and attitude, influences the behavioral intention of Generation Y to use mHealth apps during the pandemic. Aside from theoretical contributions, the findings provide practical suggestions for app developers and marketers to improve mHealth apps and services. Based on the diffusion of innovation (DOI) theory, this study addresses the following questions: 

**RQ1.** To what extent do relative advantage, compatibility, complexity, observability, and trialability of mHealth contribute to the behavioral intention of using mHealth services during the COVID-19 pandemic?

**RQ2.** What are the moderating impacts of attitude and self-discipline motivation on the relationships between trialability and behavioral intention to use mHealth apps? 

The first section of the paper describes a comprehensive review of the relevant literature and the subsequent development of a theoretical model of users’ behavioral intention based on the DOI theory. Next, the research methods and empirical findings of the study are described and followed by the results section. Finally, the theoretical contributions, implications, study limitations, and future directions for this study are presented.

## 2. Theories and Hypothesis Development

In view of mHealth as an emerging and innovative technology, a conceptual model was proposed in this study to evaluate the relationship between the perceived characteristics of innovation with the trialability and intention to use mHealth apps [36]. This model evaluates the users’ behavioral intention in terms of the trialability of mHealth by using knowledge, attitude, and self-discipline motivation as moderators to examine the underlying relationships among exogenous and endogenous variables. The DOI theory offers an appropriate framework for the analysis of factors influencing the successful user adoption rate and implementation of healthcare innovations such as mHealth apps. The DOI theory mentions five different characteristics of innovation including relative advantage, complexity, compatibility, observability, and trialability that explain how and why users make decisions based on their intention to use any new technology [40]. 

Given that the mHealth app is not entirely adopted among the users, the present study endeavors to unearth the mechanism of the behavioral intention to use the mHealth app by providing free services or trial versions among the users so that they can evaluate it before full adoption. Prior studies stated that any positive outcome from evaluating any innovative device increases commitment and validates the expectation for the final adoption [35,36]. Henceforth, among these five perceived characteristics, the present study considers the first four factors as the gateways for the trialability (i.e., evaluation or assessment tool) of the mHealth app to the behavioral adoption among Generation Y [35].

A conceptual framework is proposed based on insights gained from previous literature and theoretical underpinnings [35]. The conceptual model and study hypotheses are illustrated by the inter-relationships between the endogenous and exogenous variables using the DOI theory (Figure 1). The hypotheses of the model are described in the following sub-sections.

### 2.1. Relative Advantage

Relative advantage is “the extent to which an innovation is considered to be better than the previous idea” [36]. By contrasting and comparing related technologies with innovative ones, users can decide to adopt technologies with greater advantages offered by the new system to improve their lives. Previous studies suggest that relative advantage is one of the most dominant predictors influencing technological adoption among users [41]. The perceived relative advantage of mHealth might include societal status, economic profitability, or individual satisfaction levels. Hence, a stronger relative advantage provided by an information system is positively associated with its implementation success [42]. A cross country survey on the COVID-19 pandemic revealed that the main advantage of a health app over traditional (manual) forms of contact tracing is that the app allows for instant alerts, automatic recording, preserving patient information (a key factor in determining the success of case isolation), and contact tracing [42]. Likewise, mHealth app services provide benefits to users by offering management and innovative features which include multimedia features, portability, context-awareness, and round-the-clock accessibility [43]. Therefore, based on the theoretical descriptions and literature review, this study postulates the following hypothesis:

**H1:** Relative advantage positively influences the trialability of mHealth apps.

### 2.2. Compatibility

Compatibility is “the extent to which an innovation is considered to be consistent with the existing system and technological values, previous experiences, and needs of potential users” [36]. Higher compatibility levels lead to easier integration with existing information systems. The compatibility of an innovation was determined by the users’ views, values, lifestyles, and situations. Technological innovations having higher compatibility with the users’ social and personal status were more likely to be accepted [41]. For example, the high compatibility of mobile phones was perceived due to low effort and fitting their lifestyles, thus making it the most widely adopted technology in society in a COVID-19 context where social distancing is highly encouraged [27]. Previous studies have shown that compatibility was associated with the behavioral intention to utilize various mobile-based facilities such as mobile apps [35]. These studies indicate that perceived compatibility is one of the significant predictors of the latent users and trialability of mHealth apps. Therefore, based on the theoretical assumptions and previous research findings, we propose the following hypothesis:

**H2:** Compatibility is positively associated with the trialability of mHealth apps.

### 2.3. Complexity

Complexity is “the extent to which technology or an innovation is regarded as difficult to use as compared to its predecessor to yield equal or comparable outputs” [36]. Complexity was a factor that influenced the adoption of technological upgrades. Despite varying complexities among the apps [44], the extent of complexity deters the apps users’ trialability and intention to use [45]. As a result, these barriers associated with mHealth apps may expose users to increased health hazards such as inconsistent functionalities, delayed start of treatment, and a worse pandemic condition [46]. Based on these studies, complexity is thought to be negatively associated with the trialability of mHealth apps. Accordingly, the following hypothesis is proposed:

**H3:** Complexity negatively influences the trialability of mHealth apps.

### 2.4. Observability

Observability is “the extent to which the expected results of a particular innovation is visible and liked by other individuals”. If the positive results are visible, people are likely to engage, espouse, and express an innovation [36,41]. A previous study explained that the “early majority” users will follow suit if they can relate to those who were engaged in the change process once the innovators of the idea and early adopters have incorporated this change [36]. Comparatively, mHealth apps are considered to be observable because they can be easily downloaded from the play store used on smartphones and be steadily exposed to customers [36]. Similarly, the “BD Corona Tracer” app in Bangladesh can also be downloaded instantly through a smartphone [26]. By increasing the observability of mHealth apps to potential users, more people will be willing to test and try the app. Based on the findings of the previous literature studies, the following hypothesis is postulated: 

**H4:** Observability positively impacts the trialability of mHealth apps.

### 2.5. Trialability and Intention to Use mHealth Apps 

Trialability is “the extent to which an innovation may be experimented with on a daily basis” to assess and to determine its applicability in a distinct area before acceptance [36]. Behavioral intention refers to the extent to which a person perceives his or her willingness to use mHealth apps and services [47]. Trialability provides users with an opportunity to test an innovation based on personal expectations without any commitment or upfront costs associated with its use [48]. Lin and Bautista [35] empirically tested the trialability and participants’ intention to use mHealth apps for health literacy and the results show that trialability and intention to use mHealth apps are statistically supported. The COVID-19 pandemic presents new experiences that demand continual observations and free trials of mHealth apps as a promotional strategy and a critical step of the pre-adoption process [35]. In an observational study, online respondent-driven detection was perceived as beneficial to public health services after proper evaluation of the app during the ongoing COVID-19 pandemic [49]. Accordingly, the study revealed that user behavioral intention is the primary factor associated with the perceived effective use of mHealth services and their usefulness. The theoretical basis of diffusion studies often indicates that trialability is positively associated with the adoption of mobile health application when observed weaknesses can be minimized [50]. 

**H5:** Trialability of mHealth apps positively influences behavioral intention to use mHealth apps.

### 2.6. Moderating Role of Attitude towards Knowledge and Self-discipline Motivation

#### 2.6.1. Moderating Effect of Attitude 

The readiness of an individual to carry out a certain behavior is determined by their attitude towards it [51]. Factors such as knowledge, attitude, user experience, and self-discipline motivation have not been evaluated as moderating variables in previous technology adoption studies. Most individuals are likely to carry out a behavior if they have a positive attitude toward a technology. Similarly, attitudes toward using COVID-19 mHealth tools vary widely across the technical modalities and chronic health conditions [52]. Individuals with a positive attitude toward information systems may demonstrate greater interest if they have a positive attitude toward knowledge of mHealth apps [39]. Hence, the influence of trialability on behavioral intention to use might strengthen or weaken based on the perceived positive or negative attitudes toward COVID-19 related mHealth apps. Therefore, the following hypothesis is proposed:

**H6:** Attitude towards mHealth apps’ knowledge moderates the influence of trialability on behavioral intention to use mHealth apps.

#### 2.6.2. Moderating Effect of Self-Discipline Motivation

“Self-discipline motivation” refers to the perceived ability and determination of the users to maintain self-discipline and use healthcare information from mHealth services [53]. Users of mHealth apps with health-related goals decide to model the articulated information and monitor their health goals to be rewarded or to avoid punishment. In a recent study, authors observed that potential mHealth users undertake an initial self-judgement or assessment on their aptitude to use mHealth apps and services [54]. Based on reviews of the contact tracing app, people feel enthusiastic and comfortable adopting it to improve self-monitoring, knowledge, and engagement [26]. Similarly, users who have a high degree of self-motivation receive more information from mHealth app platforms and strive to learn how to use the new information technology, thus enhancing its perceived ease of use [38]. If users have a high level of self-discipline motivation towards mHealth apps, the perceived usefulness can increase future adoption intention [38]. In contrast, users with less self-assurance in using mHealth apps may feel that it is useless, thus choosing not to adopt mHealth apps. Therefore, this study proposes the following hypothesis:

**H7:** Self-discipline motivation moderates the impact of trialability of mHealth apps on the behavioral intention to use mHealth apps.

## 3. Research Methods

### 3.1. Research Setting

The present study followed positivistic philosophy and a deductive reasoning approach and chose quantitative methodology to conduct the analysis [55]. Following the DOI theory, the proposed study used multi-item survey measures from the prior studies. In this study, the sample population is Generation Y university and college graduates in the north wing (Dhaka North City Corporation) and south wing (Dhaka South City Corporation) areas of Dhaka. The study participants were selected from Dhaka using purposive sampling to represent both the university and college graduates because of the high respondents’ concentration in Dhaka city. In total, 463 questionnaires were distributed to postgraduate students from 14 educational institutions between January and May 2020. In total, 280 usable questionnaires were obtained with a response rate of 61%. Figure 2 (on the next page) demonstrates the recruitment flow chart.

### 3.2. Participants’ Information

After screening out ineligible respondents, 51 public university graduates (18 percent), 114 private university graduates (41 percent), and 115 national university graduates participated in the study. A total of 186 (66%) respondents were males and 94 (34%) respondents were females. More than half of the respondents were between 25 and 30 years old (62%), and the remaining respondents were between 30 and 40 years old (38%). The majority of them completed postgraduate studies (43%), while the rest had bachelor’s degrees (29%) or other degrees (28%). In this survey, participant occupation (*n* = 280) was most often listed as student (50%), along with unemployed (29%), and business (21%). The majority of the participants’ economic status was classified as middle-income (59%), followed by high-income (24%), and low-income (17%) groups.

### 3.3. Measurement Tools

Multi-item scales from previous studies were used in this study and modified with input from domain experts, academics, and health professionals in Bangladesh to ensure content validity. Based on a five-point Likert-type scale ranging from 1 (strongly disagree) to 5 (strongly agree), to predict mHealth app adoption, the respondents were asked to rate their responses for each statement on relative advantage, compatibility, trialability, complexity, and observability. Survey items were adopted from a previous study [35], which included seven statements for relative advantage, four statements for compatibility, four statements for complexity, two statements for observability, and five statements for trialability. Sample items include “Using a mobile phone gives me greater control over my health” (relative advantage), “I think that using a mobile phone for health purposes fits well with the way I like to live” (compatibility)”, “Using a mobile phone for health purposes is difficult to use” (complexity), “The results of using a mobile phone for health purposes are apparent” (observability), and “I have had ample opportunities to try various mobile health apps and services” (trialability). To measure attitude towards mHealth apps, four items adapted from Hossain, Ang, Chng, and Wong [39] were used including ‘‘I am keen to learn about and try new mobile health solutions in future”. Self-discipline motivation was assessed using a three-item scale from Gimpel, Nißen, and Görlitz [53] and included items such as “I am allowed to reward myself each time I make a small progress in achieving my health goals”. To measure the behavioral intention of using mHealth, the present study adopted four items from Cho et al. [56] such as “I intend to use mHealth in the next few days”.

### 3.4. Common Method Bias

This study used several techniques to prevent the occurrence of common method bias [57]. Firstly, the confidentiality and anonymity of the participants were assured to elicit accurate responses [58]. Secondly, we ran Harman’s single-factor test and the first factor accounted for 33.07% of the variance, which was lower than the recommended maximum value of 50% of the total variance obtained [59,60]. Thirdly, we examined multi-collinearity issues, and the estimated variance inflation factor indicated no serious concern since the highest variance inflation factor for any latent variable was below 10.00 [55]. Finally, as previously suggested by researchers [61], a correlation matrix test was performed among the constructs. The estimated results listed in Table 1 indicate that none of the values exceeded 0.90. Therefore, common method bias was not an issue in this study.

### 3.5. Analytic Technique

Partial least square-based structural equation modeling (PLS-SEM) was applied in the study design as a regression analytical tool to analyze all the results based on a two-step process. The measurement model was tested using confirmatory factor analysis (CFA), and the structural model was estimated using path analysis and a global model fitness test [62,63].

## 4. Results

### 4.1. Measurement Issues

Numerous studies have confirmed that both the discriminant validity and convergent validity can be used to estimate the validity of the proposed model. Furthermore, Cronbach alpha and composite reliability (CA) were estimated to measure the internal consistency of all the items. Table 2 indicates that Cronbach’s alpha and composite reliability scored more than 0.80 for all of the constructs, which is higher than the proposed threshold value of 0.70 [62]. Table 2 also indicates that the average variance extracted (AVE) values range from 0.626 to 0.876, while the estimated standardized factor loadings for the constructs range from 0.767 to 0.935, which are higher than the threshold level [62]. Therefore, the requirements for convergent validity and internal reliability were supported in this study. 

Moreover, the study examined the discriminant validity of each scale (Table 1) and found that the square root of a construct’s AVE is larger than its correlation with other constructs [64]. Thus, the present study has no issue with measurement.

### 4.2. Structural Model Evaluation

The structural model was evaluated using various path loadings, path coefficients, and corresponding t-values observed among the constructs. Specifically, the bootstrapping method was employed to test the hypothesized relationships between the variables. Based on the coefficient of determination, the relative advantage, compatibility, complexity, and observability of the structural model accounted for approximately 62.1% (*R*^2^) of the variance in the trialability of mHealth, while the trialability of mHealth accounted for 72.5% (*R*^2^) of the variance in behavioral intention to use mHealth apps. This study also employed the goodness of fit (*GoF*) test in equation (1) which is equivalent to the square root of the average communalities and *R*^2^ of all the endogenous variables [65]. The *GoF* of the proposed model is presented as follows:(1)$$Goodness\;of\;Fit\;\left(GoF\right)=\sqrt{Average\;CM\times Average\;R^{2}}$$
$$GoF=\sqrt{0.716\times 0.673}$$
GoF=0.695

The model’s goodness of fit values of 0.10, 0.25, and 0.36 were representative of the small, medium, and large effect sizes, respectively, in which the minimum average variance extracted of any construct must be larger than 0.50 [64,66,67]. The estimated goodness of fit and minimum average variance extracted were 0.695 (*GoF*) and 0.716 (average variance extracted) which were significantly higher than the cut-off value, thereby indicating that the model’s effect size was large [67,68]. 

The results for the hypothesis tests are shown in Table 3. The results indicate that the relationships between relative advantage and trialability of mHealth (t = 3.071, β = 0.229, *p* < 0.05), compatibility and trialability of mHealth (t = 2.796, β = 0.232, *p* < 0.05), complexity and trialability of mHealth (t = 4.776, β = -0.411, *p* < 0.05), observability and trialability of mHealth (t = 2.941, β = 0.235, *p* < 0.05), and trialability of mHealth and behavioral intention to use mHealth apps (t = 4.022, β = 0.415, *p* < 0.05) were significant. Therefore, the hypotheses H1, H2, H3, H4, and H5 were supported in this study. 

As shown in Figure 3, the conceptual model also predicted a significant portion of the variance (*R*^2^ = 0.621) in the trialability of mHealth apps. A good *R*^2^ value (0.725) was also observed for the behavioral intention to use mHealth apps. Moreover, *R*^2^ values higher than 0.20 are considered high in behavioral science studies [69].

### 4.3. Moderation Effect of Knowledge Attitude and Self-Discipline Motivation

The moderating effects of attitude and self-discipline motivation were tested using hierarchical regression analysis. Table 4 depicts the interaction effects of attitude and self-discipline motivation. Firstly, the intervening effect of attitude on the relationship between the trialability of mHealth apps and behavioral intention to use mHealth apps was tested. It was shown that the interaction effect of knowledge attitude on the relationship between the trialability of mHealth apps and the behavioral intention to use mHealth apps was insignificant (β = 0.043, *p* > 0.10). Secondly, the moderating effect of self-discipline motivation on the relationship between the trialability and behavioral intention to use mHealth apps was tested and was not found to be significant. We plotted the estimates in Figure 4 and Figure 5 and both figures asserted that the relationship between trialability of mHealth apps and behavioral intention to use mHealth apps was not influenced by the high and low presence of attitude. Figure 5 indicates that the relationship between trialability and behavioral intention to use mHealth apps was not affected by self-discipline motivation. Thus, no moderating effect of the hypothesized relationship was observed and both H6 and H7 were not supported in this study.

## 5. Discussion

In line with the research questions and objectives, we examined all the hypothesized relationships. Accordingly, the present study demonstrates that all the hypothesized direct effects had significant associations. However, the moderating effects of self-discipline motivation and attitude were not supported. The study demonstrates trialability as the connecting variable between mHealth app innovations and the respondents’ behavioral intention to use in a pandemic context. The DOI theory was applied in this study to determine the behavioral intention of users regarding mHealth services in Bangladesh during the pandemic [36]. Five perceived characteristics of innovation, namely relative advantage, complexity, compatibility, observability, and trialability were assessed in this study to determine how the perceptions of mHealth’s app characteristics predict intention to use an innovative app among users in a COVID-19 context. 

The primary factors responsible for strengthening healthcare among Generation Y in Bangladesh through the intention to adopt mHealth services were evaluated in this study. The results of this study are consistent with previous studies performed based on the DOI theory in the context of mHealth apps’ intention to use [35]. The results show that the DOI is an excellent predictive theory to determine the younger generation’s intention to utilize mHealth apps and services. The results reveal that four factors of DOI theory were significant as the antecedents of trialability, and trialability was a significant antecedent of behavioral intention to use mHealth apps. As compared to other developed countries, the younger generation in Bangladesh prefers to use innovative IT-based healthcare services, such as mHealth [32], and this observation was supported by previous studies [35,36]. Surprisingly, attitude appears to have an inconclusive influence and was not found to significantly affect the users’ behavioral intention to use mHealth. This result indicates that the attitude towards knowledge management for healthcare and service utilization is not as important as previously suggested by the literature studies [32,39].

Self-discipline motivation is a prepotent response and conscious effort that modulates one’s behavior toward a particular objective [70]. Hence, the trialability of mHealth users with higher self-discipline motivation is expected to be more significantly associated with behavioral intention to use. Surprisingly, self-discipline motivation does not appear to be a significant moderator of intention to adopt mHealth among Bangladesh’s youth. The results appear to contradict the DOI’s findings because low self-discipline motivation in young people does not alter the association between trialability and behavioral intention to use mHealth apps. However, it was their trialability that influenced actual usage. The relationship between trialability and behavioral intention to use was found to be unmodified by attitude and self-discipline motivation. Most people and patients use their phones to look up health-related information due to convenience, interactive features, portability, information availability, and low costs.

### 5.1. Theoretical Contributions

The empirical findings of this study contribute theoretically and practically to the existing literature in the context of mHealth’s intention to adopt in countries such as Bangladesh. Firstly, a conceptual model was proposed and developed to explore the causal relationships between the perceived innovation attributes of mHealth apps and their trialability, and user behavioral intention to utilize mHealth in a pandemic context. This model proposed novel causal paths between the trialability of mHealth and behavioral intention to use mHealth apps. This is the first study performed to explore the relationships between these constructs, which have been previously overlooked in existing studies in the context of strengthening mHealth services during COVID-19 in Bangladesh. Secondly, the underlying model was extended by including attitude and self-discipline motivation, which might alter the proposed impact of trialability on the behavioral intention of using mHealth services during COVID-19. Additionally, this effort augments the current theoretical studies on mHealth and exposes the primary factors affecting the intention and behavior of users adopting mHealth apps and services in developing countries. Thirdly, the study outcomes support the positive association between trialability and behavioral intention to use mHealth apps. The observations from the mHealth services among Generation Y will provide a very useful insight into developing an accurate mHealth app across the world.

### 5.2. Practical Implications

This study offers several practical contributions aiming to enhance healthcare capacity development. The results of this study reveal useful insights into the understanding of DOI constituents that influence the behavioral intention to use mHealth apps. The findings of this study will improve the healthcare quality for patients and individuals as well as offer numerous innovative business opportunities for various actors such as information communication technologies, healthcare, importers, high-tech vendors, hospitals, and telecommunication service providers. Therefore, this study investigates the influential factors that are related to the trialability of mHealth apps and behavioral intention to use mHealth apps and services among Generation Y in serious health emergencies. More specifically, the outcome of this study shows that the trialability needs more emphasis for understanding the intention of users to use mHealth apps. Thus, policymakers must shed light on the antecedents or factors affecting trialability and the impact of trialability on behavioral intention to use mHealth apps before launching any new mHealth app. The outcome of this study will help explore the factors to try and adopt mobile apps for healthcare delivery. Moreover, the health directorate should also create and promote effective public health campaigns and use free or limited trials as a pre-adoption process to promote the trialability of mHealth apps.

### 5.3. Limitations and Directions for Future Research

Despite the study’s insights, it has a few limitations. First, we collected cross-sectional data using purposive sampling that prevents the generalizability of the findings in other age groups and geographical settings. Future studies should collect longitudinal data from diverse age groups across Bangladesh that might ensure the generalization of the findings. Second, the mHealth apps’ sample population was limited to Generation Y recent university graduates from Dhaka, Bangladesh. Their high mHealth user ability may lead to different outcomes than the general population. Thus, future studies can include more ethnically and educationally diverse groups to broaden research generalization. Third, the findings of the study are based on Generation Y in Bangladesh that might not generalize to Generation Y in other cultural settings. Therefore, we suggest that future studies replicate the same research model in other cultural backgrounds to cross-validate the findings. Finally, another viable limitation of the study is that the actual use of mHealth apps is not predicted in the pandemic context. Hence, future researchers might extend the present research model with the inclusion of the actual use of mHealth apps in a multi-web survey. Additionally, the extended model can also be tested with diverse age groups in a cross-cultural setting. Finally, the study ignores the potential impacts of the apps’ functionality, such as online intervention, smartphones’ features and their playfulness, professional feedback, data security, etc., which were found significant predictors of an app’s adoption [71]. Thus, future researchers might enclose comprehensive research considering apps’ design and functionalities during an adoption process.

## 6. Conclusions

During this pandemic, the mHealth app has received increasing attention internationally because of social distancing. However, the factors influencing trialability and its subsequent behavioral intention to use mHealth apps in a moderated mechanism have not been extensively examined. To address this gap, the present study proposed an integrated model involving attitude, self-discipline motivation, and intention to use along with characteristics of innovation mentioned in the DOI theory. The proposed model was thoroughly examined in line with the study’s research questions and results show that four characteristics are significantly associated with the trialability which subsequently influence behavioral intention to use mHealth apps during the COVID-19 context. However, the moderating effects of attitude and self-discipline motivation are not statistically supported.

## Figures and Tables

**Figure 1 ijerph-19-02752-f001:**
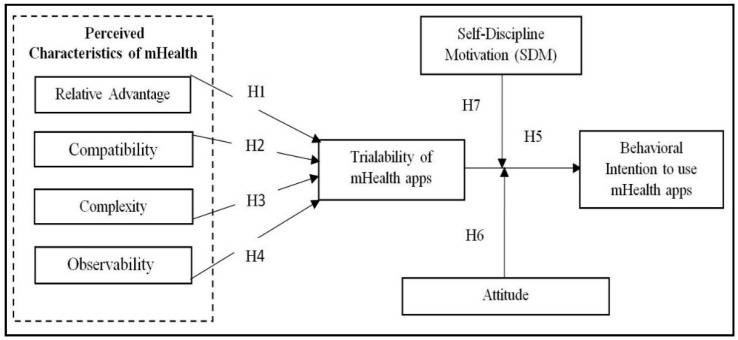
Proposed conceptual model framework.

**Figure 2 ijerph-19-02752-f002:**
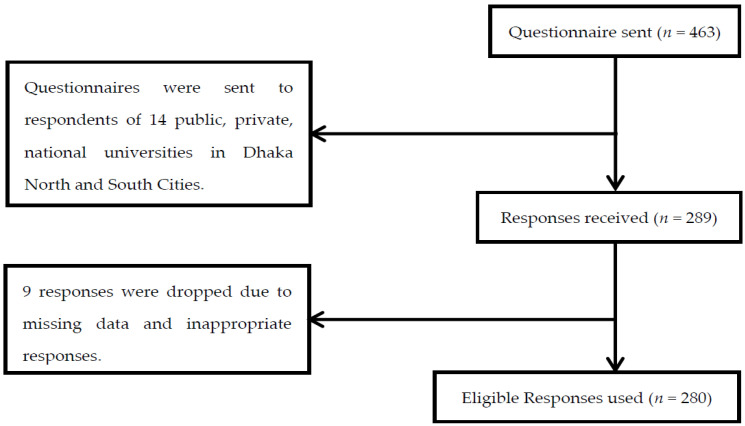
Flowchart of participants’ inclusion.

**Figure 3 ijerph-19-02752-f003:**
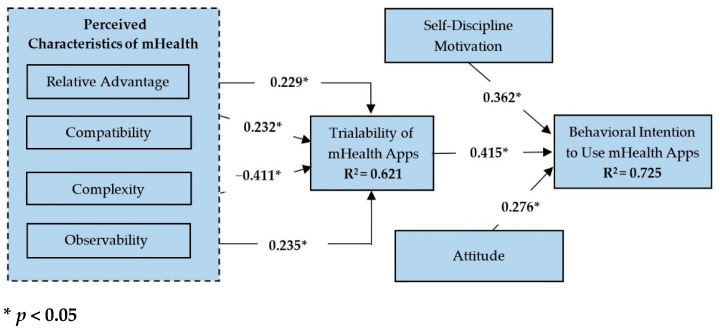
Structural model and corresponding path estimates.

**Figure 4 ijerph-19-02752-f004:**
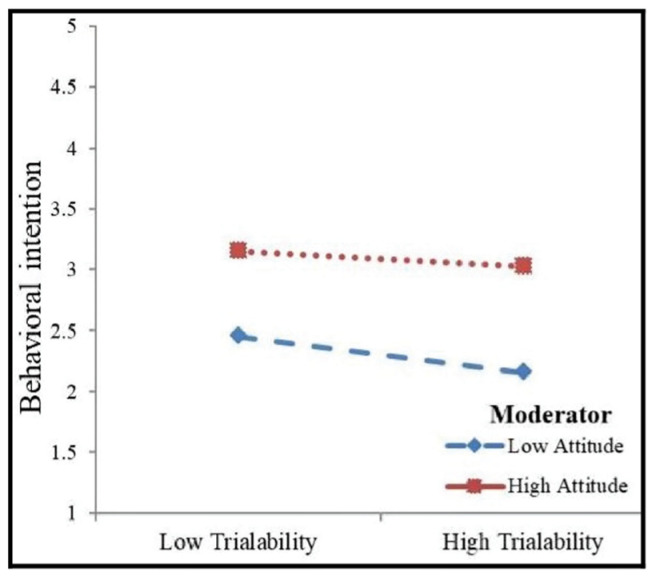
Moderating effect of attitude.

**Figure 5 ijerph-19-02752-f005:**
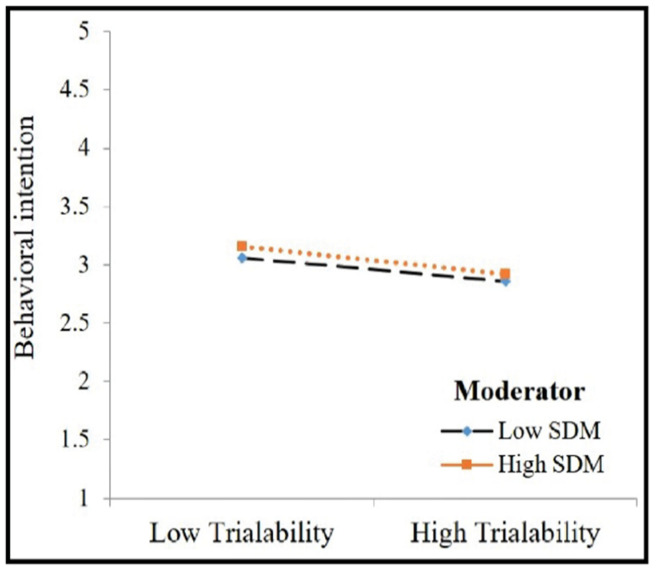
Moderating effect of SDM.

**Table 1 ijerph-19-02752-t001:** Correlation matrix estimates for discriminant validity.

Variables	1	2	3	4	5	6	7	8	9	10	11	12	13
Control variables													
1. Age	1												
2. Occupation	0.079	1											
3. Education	−0.024	0.044	1										
4. Economic status	−0.022	0.080	0.053	1									
5. Gender	0.112	0.050	0.090	−0.072	1								
Latent variables													
6. RLA	0.016	−0.083	−0.012	0.014	0.116	0.791							
7. CMP	−0.037	0.072	0.064	−0.017	0.071	0.216 **	0.869						
8. CML	−0.021	0.017	0.005	−0.103	0.062	−0.263 **	−0.484 **	0.862					
9. OBS	−0.042	0.004	−0.091	0.011	0.107	0.274 **	0.255 **	−0.354 **	0.936				
10. TLH	−0.024	0.021	0.031	−0.051	0.120 *	0.452 **	0.541 **	−0.665 **	0.503 **	0.851			
11. Attitude	0.051	0.013	0.103	−0.086	0.089	0.332 **	0.416 **	−0.502 **	0.360 **	0.724 **	0.814		
12. SDM	0.035	0.034	0.077	−0.080	0.090	0.329 **	0.323 **	−0.277 **	0.235 **	0.491 **	0.686 **	0.812	
13. BIU	−0.048	−0.032	0.032	−0.099	0.063	0.284 **	0.236 **	−0.281 **	0.191 **	0.467 **	0.629 **	0.443 **	0.825
Mean	−	−	−	−	−	3.88	3.80	1.91	3.93	3.84	3.78	3.84	3.85
Std. Deviation	−	−	−	−	−	0.575	0.818	0.755	0.865	0.749	0.657	0.623	0.598

*. Correlation is significant at *p* < 0.05 (2-tailed), **. Correlation is significant at *p* < 0.01 (2-tailed), RLA = Relative advantage, CMP = Compatibility, CML = Complexity, OBS = Observability, TLH = Trialability, SDM = Self-discipline motivation, and BIU = Behavioral intention to use.

**Table 2 ijerph-19-02752-t002:** Estimates on convergent validity and internal reliability.

Latent Variables	Items	AVE	CR	CA	SFL	t-Value
Relative advantage	RLA1	0.626	0.921	0.900	0.774	4.46
	RLA2				0.798	5.44
	RLA3				0.794	5.42
	RLA4				0.812	5.71
	RLA5				0.767	5.40
	RLA6				0.803	5.43
	RLA7				0.787	5.92
Compatibility	CMP1	0.756	0.925	0.892	0.881	10.18
	CMP2				0.871	9.44
	CMP3				0.867	9.79
	CPM4				0.858	9.45
Complexity	CML1	0.742	0.920	0.885	0.848	11.68
	CML2				0.850	11.70
	CML3				0.885	12.78
	CML4				0.864	12.98
Observability	OBS1	0.876	0.934	0.858	0.935	13.26
	OBS2				0.937	13.28
Trialability	TLH1	0.724	0.929	0.905	0.857	21.05
	TLH2				0.835	18.28
	TLH3				0.851	22.81
	TLH4				0.851	22.12
	TLH5				0.860	21.70
Attitude	AT1	0.663	0.887	0.831	0.814	9.72
	AT2				0.802	8.78
	AT3				0.811	9.12
	AT4				0.830	9.78
Self-discipline motivation	SDM1	0.660	0.853	0.743	0.807	11.93
	SDM2				0.808	11.40
	SDM3				0.822	13.46
Behavioral intention to use	BIU1	0.681	0.895	0.844	0.820	18.21
	BIU2				0.818	18.73
	BIU3				0.805	18.23
	BIU4				0.857	20.86

Note: AVE = Average variance extracted, CR = Composite reliability, CA = Cronbach’s alpha, SFL = Standard factor loadings.

**Table 3 ijerph-19-02752-t003:** Direct effects of the structural model.

Hypothesis	Path Relations	β	Standard Error	T Statistics
H1	Relative Advantage → Trialability	0.229	0.075	3.071
H2	Compatibility → Trialability	0.232	0.083	2.796
H3	Complexity → Trialability	−0.411	0.086	4.776
H4	Observability → Trialability	0.235	0.080	2.941
H5	Trialability → Behavioral Intention	0.415	0.103	4.022

**Table 4 ijerph-19-02752-t004:** Moderating effects of knowledge attitude and self-discipline motivation.

Variable	Behavioral Intention to Use		
	Model 1	Model 2	Model 3
Intercept	2.417	1.639	2.100
Trialability	0.373 *	0.020	−0.108
Attitude		0.542 *	0.391 **
Self-discipline motivation		0.021	0.038
Trialability x Attitude			0.043
Trialability x Self-discipline motivation			−0.007
*R* ^2^	0.218	0.397	0.398
Δ*R*^2^		0.179	0.001

*. Regression is significant at *p* < 0.001 and **. Regression is significant at *p* < 0.05.

## Data Availability

Not applicable.

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
