# Peer review of "Strengthening the Trialability for the Intention to Use of mHealth Apps Amidst Pandemic: A Cross-Sectional Study"

_ijerph, 2022, doi:10.3390/ijerph19052752_

Round 1

Reviewer 1 Report

This study aims to investigate the key factors influencing the trialability of mHealth apps/services and behavioral intention to adopt mobile health applications, using convenience sampling and collected responses from generation Y participants in Bangladesh. The study is scientifically sound and presented in a good manner. I therefore only have a few remarks.

Remarks:

  • line 71: "and they understand the long-term future of mHealth services" -> Is there any proof for this (incl. reference)? A few sentences later you state that 'intention to adoption is still slow'.
  • line 264-273: do you observe any differences in results between genders, age groups, education levels, occupations and income levels?

Spelling/grammar corrections:

  • line 68: "Generation-Y" -> "Generation Y".
  • line 84: "influences" -> "influence"? Depending on what it refers to exactly.
  • line 85: "Gen Y" -> first introduce the abbreviation.
  • line 195: "smart phone" -> "smartphone".
  • line 256: "generation Y" -> "Generation Y". Be consistent throughout the paper.
  • line 358: "([61]" -> "[61]".
  • line 363: "was representative" -> "were representative".
  • line 369: "PLS" -> "Partial Least Squares (PLS)".
  • figure 4 and 5: improve quality of figures.

Author Response

Response to Reviewer - 1

Enormous thanks to the reviewer for the learned comments and discerning suggestions for the revision. As per the reviewer’s suggestion, we revised/corrected the paper in the following ways:

Comment 1

This study aims to investigate the key factors influencing the trialability of mHealth apps/services and behavioral intention to adopt mobile health applications, using convenience sampling and collected responses from generation Y participants in Bangladesh. The study is scientifically sound and presented in a good manner.

Our response

Thank you so much for your comment.

Comment 2

line 71: "and they understand the long-term future of mHealth services" -> Is there any proof for this (incl. reference)? A few sentences later you state that 'intention to adoption is still slow'

Our response

Thank you so much for your comments. We eliminated these texts.

Comment 3

Line 264-273: do you observe any differences in results between genders, age groups, education levels, occupations and income levels?

Our response

Thank you so much. We have not noticed any significant effects of those variables on the latent variables.

Comment 4

Line 68: "Generation-Y" -> "Generation Y".

Our response

Thank you so much for noticing this mistake and we corrected it.

Comment 4

Line 84: "influences" -> "influence"? Depending on what it refers to exactly.

Our response

Thank you so much for noticing this mistake and we corrected it.

Comment 5

Line 85: "Gen Y" -> first introduce the abbreviation.

Our response

Thank you so much for noticing this mistake. We corrected this and used ‘Generation Y’ everywhere.

Comment 6

Line 195: "smart phone" -> "smartphone".

Our response

Thank you so much for noticing this mistake and we corrected it.

Comment 7

Line 256: "generation Y" -> "Generation Y". Be consistent throughout the paper.

Our response

Thank you so much for noticing this mistake and we corrected it.

Comment 8

Line 358: "([61]" -> "[61]".

Our response

Thank you so much for noticing this mistake and we corrected it.

Comment 9

Line 363: "was representative" -> "were representative".

Our response

Thank you so much for noticing this mistake and we corrected it.

Comment 10

line 369: "PLS" -> "Partial Least Squares (PLS)".

Our response

Thank you so much for noticing this mistake and we corrected it.

Comment 3

Figure 4 and 5: improve quality of figures.

Our response

Thank you so much for noticing this. We significantly improve the quality of these two figures.

Thank you so much for appreciating the present work. We do hope that you will find this draft acceptable now.

Sincerely

Authors.

Reviewer 2 Report

I would like to thank for the opportunity to review the manuscript "Strengthening the Trialability for the Intention to Use of mHealth Apps Amidst Pandemic: A Cross-sectional Study". Overall the manuscript shows a good adoption of the scientific method, the topic is timely, and the results are well-conducted. In general, these findings can be useful to better design the development and spread of mHealth systems that more and more important as recently shown during the COVID-19 pandemic.

I have no particular comments about the study conduction; nevertheless, I feel that the readers might benefit for a better discussion about the use of mHealth apps during this pandemic and the benefits that could arise for their optimal spread. For example, the authors could consider some of the previous literature about the use of smart devices and wearables for monitoring physical activity and health during the quarantine and lockdown periods (Buoite Stella et al., 2021; Khan et al., 2021; Chen and Wang, 2020; von Humboldt et al., 2020; Lukas et al., 2020), and how these system might also assist medical checks and promote healthy lifestyles (McGarrigle and Todd, 2020; Lee et al., 2019).

Author Response

 Response to Reviewer – 2

Comment 1

I would like to thank you for the opportunity to review the manuscript "Strengthening the Trialability for the Intention to Use of mHealth Apps Amidst Pandemic: A Cross-sectional Study". Overall the manuscript shows a good adoption of the scientific method, the topic is timely, and the results are well-conducted. In general, these findings can be useful to better design the development and spread of mHealth systems that are more and more important as recently shown during the COVID-19 pandemic.

Our response

Thank you so much for your appreciation.

Comment 2

I have no particular comments about the study conduction; nevertheless, I feel that the readers might benefit for a better discussion about the use of mHealth apps during this pandemic and the benefits that could arise for their optimal spread. For example, the authors could consider some of the previous literature about the use of smart devices and wearables for monitoring physical activity and health during the quarantine and lockdown periods (Buoite Stella et al., 2021; Khan et al., 2021; Chen and Wang, 2020; von Humboldt et al., 2020; Lukas et al., 2020), and how these system might also assist medical checks and promote healthy lifestyles (McGarrigle and Todd, 2020; Lee et al., 2019).

Our response

Thank you so much for your comments. We revised introduction section and cite those references following your comments.

Thank you so much for your constructive comments. We hope that you will find this paper acceptable now.

Sincerely,

Authors.

Reviewer 3 Report

The paper contains sufficiently new and adequate information, and it adheres to the journal’s standards. The topic and level of formality are appropriate for the journal`s readership. Its style and readability are suitable. There is a huge amount of information given throughout the article, but I would suggest revising the paper. 

Research question 2 (RQ2) must be completed. Define the core variables of the proposed model!

The methodological concept is clear. The selected methodology is scientifically appropriate, but the methods are not described in detail.

I also miss recent relevant literature in this area. I suggest citing: TOMAŽIČ, Tina, STANOJEVIĆ-JERKOVIĆ, Olivera. Online interventions for the selective prevention of illicit drug use in young drug users : exploratory study. Journal of medical internet research. April 2020, vol. 22, no. 4. DOI: 10.2196/17688.

Results are presented clearly and analyzed appropriately. Major idea received enough attention and explanation.

Theoterical and practical implications are well presented.

In summary, the article is sufficiently interesting to warrant publication, but it needs minor revision. Please follow all the comments above.

Author Response

Response to Reviewer - 3

Comment 1

The paper contains sufficiently new and adequate information, and it adheres to the journal’s standards. The topic and level of formality are appropriate for the journal`s readership. Its style and readability are suitable. There is a huge amount of information given throughout the article, but I would suggest revising the paper. 

Our response

Thank you so much for your compliments and asked us to revise. We consider each of your concerns and revised the draft accordingly.

Comment 2

Research question 2 (RQ2) must be completed. Define the core variables of the proposed model!

Our response

Thank you so much for noticing this confusion and we corrected it.

Comment 3

The selected methodology is scientifically appropriate, but the methods are not described in detail.

Our response

Thank you so much for your insightful comments to improve our draft. We paid attention to your comments and significantly improved the draft.

Comment 4

I also miss recent relevant literature in this area. I suggest citing: TOMAŽIČ, Tina, STANOJEVIĆ-JERKOVIĆ, Olivera. Online interventions for the selective prevention of illicit drug use in young drug users: exploratory study. Journal of medical internet research. April 2020, vol. 22, no. 4. DOI: 10.2196/17688.

Our response

We shortened the introduction section following your guidelines.

Comment 5

Results are presented clearly and analyzed appropriately. The major idea received enough attention and explanation.

Our response

Thank you so much for your appreciation.

Comment 6

Theoretical and practical implications are well presented.

Our response

Thank you so much for your appreciation.

Comment 7

In summary, the article is sufficiently interesting to warrant publication, but it needs minor revision. Please follow all the comments above.

Our response

Thank you for providing a constructive comment. We revised the draft following your comments.

Thank you so much respected reviewer (3). We revised the manuscript according to your suggestions. We hope that you will find this revised draft acceptable now.

Thank you.

Authors.
